# Approximating Max Function in Fully Homomorphic Encryption

Hyunjun Lee, Jina Choi and Younho Lee *

Department of Data Science, SeoulTech, Seoul 01811, Republic of Korea
* Correspondence: younholee@seoultech.ac.kr

**Abstract:** This study focuses on efficiently finding the location of the maximum value for large-scale values encrypted by the CKKS (Cheon—Kim—Kim–Song) method. To find the maximum value, $\log M + 1$ comparison operations and $\log M$ rotation operations, and $2 \log M + 3$ additions and $2 \log M + 1$ multiplications are required. However, there is no known way to find a $k$-approximate maximum value, i.e., a value with the same most significant $k$-bits as the maximum value. In this study, when the value range of all data in each slot in the ciphertext is [0, 1], we propose a method for finding all slot positions of values whose most significant $k$-bits match the maximum value. The proposed method can find all slots from the input ciphertexts where their values have the same most significant $k$-bits as the maximum value by performing $2k$ comparison operations, $(4k + 2)$ multiplications, $(6k + 2k \log M + 3)$ additions, and $2k \log M$ rotation operations. Through experiments and complexity analysis, we show that the proposed method is more efficient than the existing method of finding all locations where the $k$ MSB is equal to the maximum value. The result of this can be applied to various privacy-preserving applications in various environments, such as IoT devices.

**Keywords:** max function; fully homomorphic encryption; applied cryptography; information security





## 1. Introduction

With the increase in privacy-related regulations, such as GDPR (General Data Protection Regulation) [1,2], the importance of privacy preservation methods for data analysis and processing is gradually increasing. Among the various types of privacy preservation techniques, homomorphic encryption (HE) enables computation over encrypted data without decryption. By supporting homomorphic operations on ciphertexts, a new ciphertext can be created that has the encryption of the operation result. Moreover, HE has the advantage of not requiring intermediate interaction with the owner of data to obtain the result if these computations are outsourced.

Furthermore, the user can obtain the result by decrypting the ciphertext without additional interaction with the cloud server that performed the computation once she receives the ciphertext containing the computation result. Such convenience enables HE to be widely used as a tool to build up privacy-preserving protocols thanks to its easy usage. Although various methods have been proposed for the realization of homomorphic encryption, two methods are currently receiving the most attention. One is TFHE [3], which supports very fast bootstrapping operation, can process the underlying plaintexts in ciphertexts in bit units, and provides table lookup functions during bootstrapping thus can perform certain homomorphic computations with a ciphertext even during bootstrapping. Another is called the CKKS method which can efficiently perform multi-precision fixed point multiplication operations with complex numbers [4,5]. This method had a disadvantage in that the bootstrapping operation was slower than TFHE, but its performance has recently been improved in various ways [6–9]. However, since CKKS has to perform various logical and statistical operations only with polynomial operations, many functions should be approximated with polynomial expressions to perform them with encrypted input. For example, the comparison function [10–12], logit function [13], and round function [9] were

approximated with polynomials, and there were other attempts to replace ReLu and tanh functions for machine learning to similar polynomial functions [14].

As an extension of this flow, this study aims to efficiently calculate the location of the maximum value for large-scale values in homomorphically encrypted data by the CKKS method. If the number of slots of the ciphertext is $M$ and all slots have input data, in order to find the location of the maximum value for $M$ numbers, the maximum value must be obtained first. For this, $(\log M)$ comparison operations and rotation operations, and two addition (or subtraction) and multiplication operations are required. After that, to find the slot where the maximum value exists and set the value of that slot to 1, that is, to find the location of the maximum value, 1 comparison, 3 additions/subtractions, and 1 multiplication are required. Therefore, combining these two processes requires up to $(\log M + 1)$ comparisons, $\log M$ rotation operations, $2 \log M + 3$ additions, and finally $2 \log M + 1$ multiplications.

However, this method can only find the position of the maximum value. If we want to find the positions of approximate maximum values, i.e., all values that have the same most significant $k$-bit as the maximum value, there is no known way to do such an operation. For example, if an operation to find the maximum value is computationally expensive, it may be advantageous to efficiently find a value whose most significant $k$-bit is the same as the maximum value in order to implement the privacy preservation operation in a low-cost device such as an IoT device, where approximate computation is quite common to reduce the required computation for energy saving [15,16]. There may also be applications where it is necessary to find all the positions in the ciphertext where approximate maximum values exist. For example, as an intermediate step to find the maximum value, we can first find approximate max values to save the computational cost.

In this study, when the value range of all data existing in each slot in the ciphertext is [0, 1], we deal with how to find all the slot positions of values whose most significant $k$-bits match the maximum value. The proposed method proceeds by finding a range of maximum values. As in binary search, the range of the maximum value is determined and repeated $k$ times to make the range half at every time. As a result, we can produce a new ciphertext where we put the value 1 to only the slot positions having the same most-significant $k$-bit value as those of the maximum value in the input ciphertext. The proposed method can carry out this by performing $2k$ comparison operations, $(4k + 2)$ multiplication, $(6k + 2k \log M + 3)$ addition, and $2k \log M$ rotation operations.

The contributions of this paper are summarized as follows.

- For the first time, we propose a method for finding all slot positions of a ciphertext where the most significant $k$-bit of the values in the slots is equal to that of the maximum value in CKKS homomorphic encryption by $O(k)$ size comparison operation.
- By way of experiment, we have shown that the proposed algorithm is superior to the existing approach in terms of computational cost if certain conditions are met (see Section 5).

The rest of the paper is organized as follows: Section 2 provides some background including the notation description and a brief introduction to CKKS. Section 3 provides the related work and the problem definition, which is followed by the proposed method in Section 4. The experimental result and the complexity analysis are provided in Section 5. The paper concludes in Section 6.

## 2. Background

### 2.1. Notation

The notation used throughout this paper is given in Table 1.

**Table 1.** Notations and Conventions.

| Symbol | Meaning |
|--------|---------|
| $\vec{z}, [\![z]\!]$ | $\vec{z} = (z_0, z_1, \cdots, z_{M-1}) \in \mathbb{R}^M$ and $[\![z]\!]$ is its encryption. |
| $M$ | The number of slots in a ciphertext. |
| $[a, b]_B$ | $\{n : a \le n \le b, n \in B\}$ ($B \in \{\mathbb{Z}, \mathbb{R}\}$). $\mathbb{Z}$ can be omitted. |
| *evk,sk,pk* | evaluation key, secret key and public key. |
| $[\![z]\!][i]$ | refers to $i$-th slot of ciphertext. |
| $c, c_x, m, m_y$ | $c, c_x$ refers to ciphertexts and $m, m_y$ refers to plaintexts ($x, y$ can be any subscript). |
| $[\![a]\!]^{-1}$ | A approximate inverse vector of $[\![a]\!]$, where $([\![a]\!](i))^{-1} \approx [\![a]\!]^{-1}(i)$ for every $i \in [0, M-1]$. |
| $\vec{1}(\vec{0}, \vec{0.5})$ | A vector where every slot is 1 (0, 0.5). |
| $\vec{1}^{(i)}$ | A vector where $i$-th slot (component) has 1 and the other slots are zero. It can be used for representing a ciphertext that contains it. |
| $\vec{1}^{[a,b]}$ | A vector where the slots between $a$-th and $b$-th have 1 including the border and the other slots are zero. It can be used to represent the ciphertext containing it. If $(a > b)$, it means the zero vector where all slot values are zero. |
| $(\vec{1}^a \| \vec{0}^b)^f$ | A vector where $(i * (a + b))$-th$\sim (i * (a + b) + a - 1)$-th slots are 1 and $(i * (a + b) + a)$-th$\sim (i * (a + b) + b - 1)$-th slots are 0 for all $i \in [0, f - 1]$. It also can be used to represent the ciphertext of the vector. |

*2.2. CKKS*

The CKKS method is a popular homomorphic encryption (HE) technique that allows efficient multiplication of encrypted complex numbers [4]. Despite only providing approximate arithmetic on encrypted data, it has been widely adopted by various privacy-preserving applications due to its fast computation speed [6].

CKKS supports the following algorithms:

- KeyGen($1^\lambda$) returns *pk*, *sk*, and *evk* after taking a security parameter $\lambda$ as input.
- Enc$_{pk}(\vec{x})$ outputs $[\![x]\!]$. Every component in $\vec{x}$ places at the corresponding slot in $[\![x]\!]$.
- Dec$_{sk}([\![x]\!])$ produces $\vec{x}$ only if $[\![x]\!]$ is a valid encryption from $\vec{x}$, which is a result of Enc or created through a set of operations with valid ciphertexts with correct *pk* and *evk*, where *pk* and *evk* are mapped to *sk*. Otherwise, it returns $\perp$.
- Add($[\![x]\!], [\![y]\!]$)(Sub($[\![x]\!], [\![y]\!]$)) outputs a new ciphertext $c$ that is an encryption of $\vec{x} + \vec{y}$ ($\vec{x} - \vec{y}$). We denote it as $[\![x]\!] \boxplus [\![y]\!]$ ($[\![x]\!] \boxminus [\![y]\!]$) to simplify the description.
- Add($[\![x]\!], k$)(Sub($[\![x]\!], k$)) outputs a new ciphertext, which represents the encryption of $(x_0 + k, \cdots, x_{M-1} + k)$ ($(x_0 - k, \cdots, x_{M-1} - k)$) for given $k \in \mathbb{C}$. To make the description simpler, it is represented as $[\![x]\!] \boxplus k$ ($[\![x]\!] \boxminus k$).
- Level($[\![x]\!]$) returns the level of $[\![x]\!]$'s, which is a non-negative integer representing the number of further possible multiplication with the ciphertext $[\![x]\!]$.
- Mult$_{evk}([\![x]\!], [\![y]\!]$) provides an (approximate) encryption of $(x_0 * y_0, \cdots, x_{M-1} * y_{M-1})$ with a level of Min(Level($[\![x]\!]$), Level($[\![y]\!]$)) $- 1$. For the purpose of simplifying the description, it is referred to as $[\![x]\!] \boxdot \mathbf{y}$.
- Mult$_{evk}([\![x]\!], k$) outputs a $c$ that is an encryption of $(kv_0, \cdots, kv_{M-1})$ where $k \in \mathbb{C}$. The level of $c$ is reduced from the level of $[\![x]\!]$ by 1. To simplify the notation, it is referred to as $[\![x]\!] \boxdot k$.
- Rot$_{evk}([\![x]\!], i$) returns an encryption of $(x_i, x_{i+1}, \cdots, x_{M-1}, x_0, \cdots, x_{i-1})$, where $i \in [0, M-1]$. If $i \in [-(M-1), -1]$, we set $i = i + M$ to make $i \in [0, M-1]$.
- Boot$_{evk}([\![x]\!]$) returns a new ciphertext $c'$ that has approximation of $\vec{x}$ if Level($[\![x]\!]$) $\ge l_{min_{boot}}$, the number of required multiplication level to perform Boot. $l_{min_{boot}}$ depends on what bootstrapping algorithm is used and parameter.

It is assumed that the rescaling algorithm [4] is carried out within the Mult algorithm. The details of the keys used have been omitted for simplicity. If multiplication requires bootstrapping, it is assumed to be carried out automatically, and the related part has been omitted for clarity of the algorithm description. Furthermore, the RNS-CKKS implementation, which leverages the GPU for improved performance [5–7,17], is used.

The parameters for CKKS are as follows: the number of slots ($M$) is 32,768, up to 9 multiplications are permitted between bootstrapping operations, the initial multiplication depth before the first bootstrapping is 21, and the value of $l_{min_{boot}}$ is 3.

The method in [11], referred to as $\mathsf{ApproxSign}(\llbracket x \rrbracket)$, is used. It takes a single ciphertext $\llbracket x \rrbracket$ and returns the encryption of a vector $(a_0, \cdots, a_{M-1})$, with $a_i$ set to 1 if $\llbracket x \rrbracket[i] > 0$, $a_i$ set to 0 if $\llbracket x \rrbracket[i] = 0$, or $a_i$ set to -1 otherwise ($i \in [0, M-1]$).

## 3. Related Work and Problem Definition

Concentrating on CKKS-based research, the first research related to the max function is a study on [10], which studied the comparison of elements in two ciphertexts of the CKKS method. More specifically, the authors in [10] focused on calculating the sign function. By subtracting one input from another and using the result of the subtraction as an input of the sign function, the sign function can be used for comparison. Since then, there have been studies [11,12] that have improved the efficiency and accuracy of homomorphic sign functions. Independently of this, there was a study that can calculate a step function that can be used for comparison on encrypted input [18].

Apart from homomorphic comparison, there are some studies that focus on efficient data sorting on the encrypted data by CKKS [19].

Unfortunately, there is no research that finds the positions (slots) of all the values with the same most significant $k$-bit as the maximum value. This study aims to efficiently find such functions in CKKS homomorphic encrypted data.

The specific problems to be addressed in this study are as follows. Let $M$ be the maximum number of data that can fit in one ciphertext in CKKS homomorphic encryption. Then, among these $M$ numbers, there will be a maximum value. Our goal is to simultaneously find the positions of slots where these maximum values and values with the same most significant $k$-bits exist. However, due to the characteristics of CKKS homomorphic encryption, the plaintext range is limited to [0, 1], which is the plaintext range in which all homomorphic operations can be easily performed including various polynomial approximation functions and bootstrapping. The goal of this study is to find an efficient algorithm that can achieve the goal, where the term 'efficient' means that the algorithm can complete its task in the linear complexity to $k$ (thus logarithmic to the entire input space).

## 4. Proposed Method

In this section, the proposed method is described. To help understand the proposed method, an overview is given first, and then the detailed process is explained using pseudo-code.

### 4.1. Overview

There are two techniques used in the proposed method. For example, when there is a cipher text $c$ containing input values, it is possible to find out whether the input value present in each slot is greater than $1/2$ by performing $\mathsf{ApproxSign}(c - 1/2)$. The $\mathsf{ApproxSign}()$ function memorizes the position of slots whose values are greater than 0 in the input ciphertext, then sets 1 to the slots at the same position in the resulting ciphertext. Similarly, it sets 0 in the output ciphertext for the slots with a value of 0 in the input ciphertext and set it to $-1$ if it has a negative value.

We use the result ciphertext of $\mathsf{ApproxSign}(c - 1/2)$ to find the range of the maximum value. If there is even one slot whose result is 1, the maximum value among the values in the input ciphertext is at $(1/2.1]$, otherwise it is in $[0.1/2]$. Hence, by using this fact, if the result value of $\mathsf{ApproxSign}(c - 1/2)$ is less than 1 in all slots, the value that is



subtracted from c changes from $1/2$ to $1/4$ and runs $\text{ApproxSign}(c - \vec{1/4})$ again. If not, run $\text{ApproxSign}(c - \vec{3/4})$ to proceed to the next step. By repeating this routine $k$ times, we can obtain the result.

We discuss why the iterations of the above steps by $k$ times can find slot locations where values with the most significant $k$-bit equal to the maximum value exist. What we need to explain is the meaning of the result of $\text{ApproxSign}(c - \vec{1/2})$. If one or more slots representing 1 are found in the resulting ciphertext, it means that the most significant bit of the maximum value in the ciphertext of c is 1. (Note that it is assumed that the input range is [0, 1]) Therefore, the slot positions of inputs with the same most significant bit as the maximum value can be found by executing $\text{ApproxSign}(c - \vec{1/2})$. Of course, there may not be a value greater than $1/2$ in any slot in $c$. In this case, every slot in the resultant ciphertext of $\text{ApproxSign}()$ returns $-1$. In this case, we can conclude that the most significant bit is zero. By repeating the same procedure with $1/4$ (or $3/4$ depending on the result of the previous $\text{ApproxSign}()$), we can find the positions of the values whose most significant two bits are the same as the maximum value. If we repeat this procedure by $k$-times, we can find the slot positions of the values which have the same most significant $k$-bit as the maximum value.

### 4.2. Description of the Proposed Algorithm

This subsection deals with the details of the proposed method. The proposed method is described below.

- Input: ciphertext $c$ containing all input data in $M$ slots, bit value (integer) $k$.
- Output: Cipher text $c_{\text{out}}$ with 1's in slots containing values with the same most significant $k$-bit value as the maximum value and 0's in other positions.

The procedure for the proposed method is given as follows:

---

**Algorithm 1** k-approximate max algorithm

---

1: **procedure** k-ApproxMax$(c, k)$
2:     $i \leftarrow 0, c_s \leftarrow [\![\vec{0.5}]\!]$
3:     $c_0 \leftarrow \text{ApproxSign}(c - c_s)$
4:     $c_0' \leftarrow (c_0 \boxplus \vec{1}) \boxdot \vec{0.5}$
5:     $c_1 \leftarrow$ (Sum up all slot values in $c_0$ and put the result into 0-th slot)
6:     $c_1 \leftarrow$ (Copy $c_1[0]$ to every slot in the ciphertext)
7:     $c_2 \leftarrow \text{ApproxSign}(c_1 \boxdot \frac{\vec{1}}{M})$
8:     $i \leftarrow i + 1$
9:     **if** $i == k$ **then**
10:         $c_2' \leftarrow (c_2 \boxplus \vec{1}) \boxdot \vec{0.5}$
11:         $c_3 \leftarrow c_2' - \vec{1}$
12:         $c_4 \leftarrow c_0' - \vec{1}$
13:         $c_{out} \leftarrow -((c_4 \boxdot c_2') \boxplus (c_0' \boxdot c_3))$
14:         **return** $c_{\text{out}}$
15:     $c_{tmp} \leftarrow (2c_2 \boxminus 1) \boxdot (\vec{0.5}^i)$
16:     $c_s \leftarrow c_s \boxplus c_{tmp}$
17:     Goto 3.

---

### 4.3. Subroutines

In the description of the Algorithm 1, which is the proposed method in the previous subsection, Step 4 and Step 5 are not explained in pseudo-code, but in text. Therefore, in order to implement the proposed method, the corresponding steps must be described in detail using pseudo-code. This subsection describes how to implement the two processes.

First, Step 4 is described. The input of this stage is the ciphertext $c_0$ where every slot has a value, and the result value is the sum of the values of all slots of $c_0$. It is stored in the 0th slot of the output ciphertext $c_1$ ($c_1[0]$), and the remaining slots of $c_1$ are filled with zeros.

1.   $j \leftarrow 1$
2.   While ($j < \log M$) do

    (a)    $c_{tmp} \leftarrow \text{Rot}(c_0, -j)$
    (b)    $c_0 \leftarrow c_0 \boxplus c_{tmp}$
    (c)    $j \leftarrow j * 2$

3.   $c_1 \leftarrow c_0 \boxdot \vec{1}^{(0)}$
4.   Return $c_1$

We proceed to discuss Step 5. Step 5 is an algorithm that copies the value of $c_1[0]$ to all slots in the resulting ciphertext and creates the resulting ciphertext so that all slots have the same value of $c_1[0]$. The process is given as follows.

1.   $j \leftarrow 1$
2.   While ($j < \log M$) do

    (a)    $c_{tmp} \leftarrow \text{Rot}(c_0, j)$
    (b)    $c_0 \leftarrow c_0 \boxplus c_{tmp}$
    (c)    $j \leftarrow j * 2$

3.   $c_1 \leftarrow c_0 \boxdot \vec{1}^{(0)}$
4.   Return $c_1$

## 5. Experimental Results and Complexity Analysis

In this section, the experimental results and computational complexity analysis of the proposed method and the existing method are discussed.

### 5.1. Experimental Result

First, the experimental results are described. The CKKS parameters are as follows. The number of slots = 32,768, log(QP) = 1555, secret key hamming weight = 192 bit, the depth of multiplication up to the first bootstrapping after encryption is 21, and 9 multiplications are possible between the subsequent bootstrapping. The plaintext bit-precision provided by the ciphertext is 42 bits.

Using these CKKS parameters, the HEaaN GPU library [20] was used to conduct experiments in the following environment: CPU = AMD 3950X, RAM = 128 GB, GPU = Nvidia RTX 6000.

The execution time of each unit operation provided by the HEaaN GPU library is as follows [20]. We measured them by 100 times and the result is given in Table 2.

**Table 2.** Average Time (ms) of CKKS unit operations and subroutines.

| Add | Mult (lv.9) | Mult (lv.1) | Rot | Boot | ApproxSign() |
|---|---|---|---|---|---|
| 0.060 | 0.864 | 0.417 | 0.776 | 115.7 | 405.0 |

The performance of the proposed algorithm based on these basic operations is as follows. Table 3 shows the average and standard deviation measured after running 10 times in each case. From the table, we can see that the running time increases linearly in terms of $k$.

**Table 3.** Average and standard deviation of running time of the proposed algorithm (in ms).

| $k$ | 1 | 2 | 3 | 4 | 5 | 6 | 7 | 8 | 9 | 10 |
|---|---|---|---|---|---|---|---|---|---|---|
| AVG | 1080.2 | 1964.1 | 2734.1 | 3507.5 | 4281.3 | 5055.0 | 5827.2 | 6602.4 | 7376.4 | 8152.8 |
| STDEV | 0.4472 | 0.4868 | 1.5189 | 2.9761 | 1.0289 | 2.2819 | 0.8204 | 1.8407 | 1.7393 | 2.2627 |

### 5.2. Computational Complexity Analysis

In this subsection, a comparison of computational complexity between the proposed method and a well-known basic method for finding the maximum value in CKKS is performed. The comparison results are described in Table 4 below.

**Table 4.** Computational complexity comparison between the proposed method and the basic max function.

|  | Computation Cost |
|---|---|
| Basic max function | $(\log M + 1)\text{ApproxSign}() + (2\log M + 3)\text{Add} + \log M\text{Rot} + (2\log M + 1)\text{Mult}$ |
| Proposed method | $2k\text{ApproxSign}() + (6k + 2k\log M + 3)\text{Add} + 2k\log M\text{Rot} + (4k + 2)\text{Mult}$ |

To analyze the results in Table 4, let us first look at the number of instances of ApproxSign() and Mult, which consume the most amount of computation. The proposed method consumes an amount of computation linearly proportional to $k$, and the conventional method for finding the position of the maximum value requires an amount of computation proportional to $\log M$. In practice, the proposed method finds multiple values, not just one. Therefore, the above comparison is not fair, and the cost for the max function that is actually compared must be multiplied by a certain constant. If we say it is $q$, the existing method requires $q(\log M + 1)$ of ApproxSign() and $q(2\log M + 1)$ of Mult. In contrast, the proposed method requires $2k\text{ApproxSign}()$ and $(4k + 2)$ Mult, so the computational complexity actually varies depending on the values of $q$, $\log M$, and $k$.

Based on this, if we think that the input data is a value sampled from the uniform distribution in (0, 1), for a fixed $k$, the positions of about $M/2^k$ slots are returned if we run the proposed method. If we set this value as $q$ and consider the most expensive operation ApproxSign(), we can conclude that if $k + \log k + 1 < (\log M)(\log M + 1)$, the proposed method is superior to the existing method in terms of the computational cost. This inequality is from $2k < M(\log M + 1)/2^k$.

## 6. Conclusions and Future Work

In this study, among the values located in all slots of one ciphertext, we discussed how to generate a new ciphertext with a value of 1 only at the position of the slot containing the value whose most significant k-bit is equal to the maximum value. This method was shown to be more efficient in terms of size comparison operation and homomorphic multiplication operation, which require the most computational cost, compared to the existing method of finding the position of the maximum value. The proposed method can be applied to applications where the exact maximum value or exact top-$k$ values are not required but the positions of the approximate maximum values are sufficiently good to be used instead.

One remarkable thing in this study is that, unlike many homomorphic computation algorithms whose complexity is linear, even their plaintext versions have logarithmic complexity, and the proposed homomorphic algorithm preserves the logarithmic complexity. As a future study, we must explore what characteristics must be present to maintain efficiency in an algorithm of plaintext input if we convert it to the homomorphic version of the algorithm.

**Author Contributions:** Conceptualization, Y.L.; methodology, Y.L.; software, H.L. and J.C.; validation, Y.L.; writing—original draft preparation, Y.L.; writing—review and editing, Y.L.; supervision, Y.L.; funding acquisition, Y.L. All authors have read and agreed to the published version of the manuscript.

**Funding:** This study was supported by the Research Program funded by the SeoulTech (Seoul National University of Science and Technology).

**Data Availability Statement:** No new data were created or analyzed in this study. Data sharing is not applicable to this article.

**Conflicts of Interest:** The authors declare no conflict of interest. The funders had no role in the design of the study; in the collection, analyses, or interpretation of data; in the writing of the manuscript; or in the decision to publish the results.

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
