# Peer review of "Approximating Max Function in Fully Homomorphic Encryption"

_electronics, doi:10.3390/electronics12071724_

Round 1

Reviewer 1 Report

In this study, the authors suggest a method to find the maximum value and approximate value from data that are encrypted by the CKKS method. The explanation of the implemented method has already been well presented, and an appropriate experiment was set up to demonstrate the soundness of the provided method. But there are some points that need to be improved. Firstly, the main contribution of this paper is providing a method for finding the k-approximate maximum value, but the necessaries of this needed to be better presented. Secondly, in the Abstract and Introduction, the authors state that the provided method is appropriate for IoT devices, but the experimental results are insufficient to support this statement. Finally, they need a comparison with the existing works to reveal the efficiency of the provided method with the existing method.

Author Response

Dear reviewer #1, 

Following your comment, we added two references [16,17] in line #61 of the paper to explain the approximate computing is performed in IoT environment. Also, approximate function can be used where the exact computation is costly [16,17]. Also, we would like to emphasize that this is the first construction of homomorphic approximate-k max function in the literature. 

Thank you.

-Younho Lee

Reviewer 2 Report

·         Some words were not written in full at 1st usage before using them as acronyms , words such as CKKS, TFHE

·         Under section 5, when discussing the result, the author should refer to the specific table by using table number like table 1, table 2 etc  i.e from this statement , “The performance of the proposed algorithm based on these basic operations is as 245 follows. The table shows the average and standard deviation measured after running 10 246 times in each case. From the table, we can see that the running time increases linearly in 247 terms of k.” which table ?

Author Response

Dear reviewer #2,

Following your comments, we have modified our manuscript as follows.

1) We added the full descriptions of the acronyms in page #1. E.g. CKKS (Cheon-Kim-Kim-Song) and GDPR (General Data Protection Regulation). 

2) We modified #247 and #249 of the paper to clarify which table we are dealing with.

Thank you.

-Younho Lee

Reviewer 3 Report

The authors provide a method to approximate max function in fully homomorphic encryption, reporting the runtime of the proposed algorithm on a custom setup. The contributions of this document are very interesting. Nevertheless, I recommend the following suggestions to improve the quality of the document. 

  1. In the introduction the authors said “The result of this can be applied to various privacy-preserving applications on various environments as IoT devices”. However, the results of the paper do not demonstrate any information about the reliability of the proposed method in an IoT device.

  2. The document has some abbreviations that are not defined, increasing the complexity for the reader. Therefore, I recommend that the authors define at least the next abbreviations: GDPR, TFHE, and CKKS.

  3. In the introduction, the authors write the next phrase “There may also be applications where it is necessary to find all the positions in the ciphertext where approximate maximum values exist”. I recommend citing an example or providing the names of the applications to increase the relevance of this research. 

  4. In section 4, the authors exhibit the procedure and subroutines proposed without a table. I recommend using a table using an algorithm library in latex, organizing the method presented in this paper.

  5. In table 3, the authors report the runtime of the proposed method. Then, table 4 compares the computational complexity of the basic max function and the proposed method. The authors need to implement the basic max function in the same environment to demonstrate the contribution of this document.

  6. Finally, I request a proofreading process to mitigate some grammar typos in the document. 

Author Response

Dear reviewer #3,

Following your comments, we have revised our manuscript as follows:

  1. In the introduction the authors said “The result of this can be applied to various privacy-preserving applications on various environments as IoT devices”. However, the results of the paper do not demonstrate any information about the reliability of the proposed method in an IoT device.

      ==> We added two references [16,17] in line #61 of the manuscript to indicate  that IoT devices perform approximate computing if the computational cost for exact computation is huge to save the consumed energy. 

  1. The document has some abbreviations that are not defined, increasing the complexity for the reader. Therefore, I recommend that the authors define at least the next abbreviations: GDPR, TFHE, and CKKS.

==> We provide the full descriptions of the abbreviations for readers in page #1. E.g. CKKS (Cheon-Kim-Kim-Song), GDPR (General Data Protection Regulation)

  1. In the introduction, the authors write the next phrase “There may also be applications where it is necessary to find all the positions in the ciphertext where approximate maximum values exist”. I recommend citing an example or providing the names of the applications to increase the relevance of this research. 

==> We added lines #63 and #64 to deal with this issue. 

  1. In section 4, the authors exhibit the procedure and subroutines proposed without a table. I recommend using a table using an algorithm library in latex, organizing the method presented in this paper.

==> We added Algorithm 1 in page #5 following your comment.

  1. In table 3, the authors report the runtime of the proposed method. Then, table 4 compares the computational complexity of the basic max function and the proposed method. The authors need to implement the basic max function in the same environment to demonstrate the contribution of this document.

   ==> It can be calculated as the performance of the basic operations are given in Table 2. Since M=32768 in our setting, the max function takes around 6800~7200 ms considering the number of bootstrapping operations. However, as stated in line #246~#250, it is unfair comparison as the proposed produces multiple numbers as an output.

  1. Finally, I request a proofreading process to mitigate some grammar typos in the document. 

==> We checked the typos 

Thank you.

-Younho Lee

Round 2

Reviewer 1 Report

The authors have already corrected the manuscript. It now satisfies my recommendation. I suggest this manuscript be publicized.

Reviewer 3 Report

The quality of the document improve the necessary to be published.